# Distribution of Duffy Phenotypes among *Plasmodium vivax* Infections in Sudan

**DOI:** 10.3390/genes10060437

**Published:** 2019-06-08

**Authors:** Musab M.A. Albsheer, Kareen Pestana, Safaa Ahmed, Mohammed Elfaki, Eiman Gamil, Salma M. Ahmed, Muntaser E. Ibrahim, Ahmed M. Musa, Eugenia Lo, Muzamil M. Abdel Hamid

**Affiliations:** 1Institute of Endemic Diseases, University of Khartoum, Khartoum 11111, Sudan; musabali39@yahoo.com (M.M.A.A.); safsafammo@hotmail.com (S.A.); alfaki_4@hotmail.com (M.E.); Look_like121@hotmail.com (E.G.); salmamed13@yahoo.com (S.M.A.); mibrahim@iend.org (M.E.I.); musaam2003@yahoo.co.uk (A.M.M.); 2Faculty of Laboratory Sciences, University of Sinnar, Khartoum 11111, Sudan; 3Department of Biological Sciences, University of North Carolina at Charlotte, Charlotte, NC 28223, USA; kpestana@uncc.edu; 4Faculty of Medicine, Sudan University of Science and Technology, Khartoum 11111, Sudan

**Keywords:** *Plasmodium vivax*, malaria, Duffy-negative, parasitemia, demography, Sudan

## Abstract

Negative Duffy expression on the surface of human red blood cells was believed to be a barrier for *Plasmodium vivax* infection in most Africans. However, *P. vivax* has been demonstrated to infect Duffy-negative individuals in several Central and East African countries. In this study, we investigated the distribution of Duffy blood group phenotypes with regard to *P. vivax* infection and parasitemia in Sudan. Out of 992 microscopic-positive malaria samples, 190 were identified as *P. vivax* positive infections. Among them, 186 were *P. vivax* mono-infections and 4 were mixed *P. vivax* and *Plasmodium falciparum* infections. A subset of 77 samples was estimated with parasitemia by quantitative real-time PCR. Duffy codons were sequenced from the 190 *P. vivax* positive samples. We found that the Duffy Fy(a-b+) phenotype was the most prevalent, accounting for 67.9% of all *P. vivax* infections, while homozygous Duffy-negative Fy(a-b-) accounted for 17.9% of the *P. vivax* infections. The prevalence of infection in Fy(a-b+) and Fy(a+b-)were significantly higher than Fy(a-b-) phenotypes (*p* = 0.01 and *p* < 0.01, respectively). A significantly low proportion of *P. vivax* infection was observed in Duffy negative individuals Fy(a-b-). This study highlights the prevalence of *P. vivax* in Duffy-negatives in Sudan and indicates low parasitemia among the Duffy-negative individuals.

## 1. Introduction

Malaria is a serious health problem, especially in developing countries. According to the latest estimates from the World Health Organization, there were 214 million new cases and 435,000 deaths, mostly occurring in sub-Saharan Africa [1].The predominant malaria parasite species is *Plasmodium falciparum*. However, *Plasmodium vivax* is widely spread and it has been increasingly recognized as a cause of severe malaria and even death [1,2]. Although the prevalence of *P. vivax* was believed to be much lower in Africa, using a compiled database, a recent study has highlighted the widespread presence of *P. vivax* across all malaria-endemic regions of Africa [3].

Based on the WHO World Malaria Report 2018, malaria constitutes a major public health problem with 720,819 reported confirmed cases and 3885 deaths in Sudan [1]. Most cases are caused by *P. falciparum.* However, in recent years the prevalence of *P. vivax* malaria has witnessed a sharp increase and accounted for up to one fifth of the total malaria cases [1] and disease is becoming widespread in areas where it has not been documented before and some severe cases due *P. vivax* have been reported [4,5].

The human Duffy antigen also known as Duffy antigen receptor for chemokines (DARC) or the *FY* gene is the key molecule required for *P. vivax* and *Plasmodium knowlesi* parasites to invade red cells through interaction with *P. vivax* Duffy binding protein (PvDBP) [6,7,8]. The *FY* gene is located on chromosome 1 and possesses four main allelic forms, including *FYA*, *FYB*, *FYA*^ES^, and *FYB*^ES^. This gene encodes a glycosylated membrane protein (*FY* glycoprotein) [9].

Polymorphisms in the *DARC* gene alter the expression of Duffy antigen protein and determine the Duffy blood group system [10]. The theoretical expectation is that Duffy-negative individuals lack Duffy antigen expression on their red blood cells and thus are protected against *P. vivax* infection [7]. The *FY* gene encodes red blood group antigens and has two co-dominant alleles *(FYA* and *FYB*), which differ by a single nucleotide polymorphism, 125 G→A, leading to an amino acid change from Gly to Asp at position 42. The composition of these two alleles gives rise to the Duffy phenotypes, Fy(a+b+), Fy(a+b−), and Fy(a−b+). Individuals with these phenotypes exhibit Duffy antigen expression and are considered as Duffy positive. The silent *FYB*^ES^ allele results from a mutation in the GATA box motif of the *FYB* promoter (−33T→C) which is known to disrupt the binding of erythroid transcription factor to GATA-1, leading to a reduced Duffy antigen expression [11]. The Duffy-negative homozygote with phenotype Fy(a-b-) has been shown to be most prevalent among Africans and individuals of African ancestry, who were thought to be protected from *P. vivax* infection [12]. However, recently, *P. vivax* infections in Duffy-negative individuals have been documented in several African countries [12], including Madagascar [13], Mauritania [14], Ethiopia [15], Cameroon [16], and Sudan [4], and thus Duffy negativity alone does not prevent infection with *P. vivax*. There is yet no documented evidence for the associations between Duffy phenotypes and the prevalence of *P. vivax* infection in malaria endemic regions in East Africa. The aim of this study was to determine the distribution of Duffy phenotypes and to estimate *P. vivax* parasitemia in relation to the different Duffy phenotypes.

## 2. Materials and Methods

### 2.1. Ethical Statement

The study was approved by the Research Ethical Review Committee, Institute of Endemic Diseases University of Khartoum, Sudan (certificate reference number 9/2016). Prior to enrollment in the study, written informed consent was obtained from all adult patients and parents or legal guardians of the children.

### 2.2. Study Sites and Subjects

Malaria transmission in Sudan is seasonal with two peaks of transmission: One during the rainy season between August and October and the other one during the winter season between December to February. *Anopheles arabiensis* is the principal malaria vector in Sudan [17]. Sample collection was conducted in the Central, Northern, and Eastern part of Sudan including Khartoum (15°53’11.2″N 32°31’39.9″E), New Halfa (15°20’00.0″N, 35°35’00″E), and River Nile (16°59’58.5″N 35°29’04.4″E), between August 2013 and 2017. Symptomatic malaria patients were recruited from Gerri and Gezira Slanj primary health centers in Northern Khartoum, New Halfa teaching hospital in New Halfa, as well as Seedon, Al-Zaidab, and Atabra primary health centers in River Nile. Thick and thin blood films were collected and stained with Giemsa using standard procedures. Slides were examined twice at the study centers and re-examined at the Department of Parasitology and Medical Entomology, Institute of Endemic Diseases, University of Khartoum. Microscopy positive samples were confirmed by 18S rRNA nested PCR assay. In this study, a total of 190 PCR-confirmed *P. vivax* blood samples were collected from symptomatic malaria patients (101 from Khartoum, 40 from New Halfa, and 49 from River Nile), 186 samples were confirmed *P. vivax* mono-infection and four samples were mixed infection (*P. vivax* and *P. falciparum*) based on nested PCR assay. The male to female ratio was 1.73 and the mean age was 25 years. Non-malaria subjects attending the same health facility who tested negative for *Plasmodium* by microscopy and PCR were included as controls. Matching cases were selected according to age, gender, and ethnicity.

### 2.3. Sample Collection and DNA Extraction

Approximately 3 to 4 spots of finger-prick blood samples (each 50 microliters) were collected on Whatman filter paper (Whatman International Ltd, Maidstone, England) from microscopy positive malaria patients (*N* = 190). In addition, 2 to 3 mL of whole blood was collected in EDTA tubes from 67 healthy subjects with no prior history of malaria and were tested as negative for malaria by microscopy and nested PCR at the time of blood collection. Genomic DNA was extracted from 2 to 3 dried blood spots using a Qiagen mini DNA kit (Qiagen, Germany) with minor modification, while DNA from blood samples was extracted using a DNeasy Blood kit (Qiagen, Germany). DNA was eluted in total of 40µL of warm elution buffer (AE) (Qiagen, Germany) and kept at −20 °C until used.

### 2.4. Molecular Identification of Plasmodium Species

Identification of *Plasmodium* species was performed using nested PCR. Species-specific primers that amplify the 18S rRNA gene for *P. falciparum, P. vivax, Plasmodium malariae*, and *Plasmodium ovale* as a single or mixed infection were used as described by previous studies [18]. The 18S rRNA amplicon was sequenced in both the 5’ and 3’ directions to confirm *P. vivax* infection (Macrogen Inc., Amsterdam, The Netherlands).

### 2.5. Estimation of P. vivax Parasitemia

The estimation of parasitemia was performed using the 18S rRNA SYBR green quantitative real-time PCR method. The *P.vivax* 18S rRNA specific primers (forward: 5′-GAATTTTCTCTTCGGAGTTTATTCTTAGATTGC-3′ and reverse: 5′-GCCGCAAGCTCCACGCCTGGTGGTGC-3′) were used for amplifications. Amplification was performed in 20 μL reactions which contained 10 μL of 2× SYBR Green Master Mix (Thermo Scientific, Waltham, MA), 0.5 μM of the forward and reverse primers, 2 μL of genomic DNA, and 7 μL of water. The initial denaturation was run at 95 °C for 3 minutes, followed by 45 cycles of 94 °C for 30 seconds, 55 °C for 30 seconds, and 68 °C for 1 minute which was followed by a hold step of 95 °C for 10 seconds. This was followed by a melting curve stage from 65 to 95 °C with 0.5 increments for 5 seconds [19]. Each assay included positive controls of *P. vivax* Pakchong (MRA-342G) and Nicaragua (MRA-340G) isolates, in addition to negative controls, including uninfected samples and water. A standard curve was produced from a 10-fold dilution series of the control plasmids to determine the amplification efficiency. A cut-off threshold of 0.02 fluorescence units that robustly represented the threshold cycle at the log-linear phase of the amplification and above the background noise was set to determine the *Ct* value for each assay. The mean *Ct* value was calculated from three independent assays of each sample. Samples yielding *Ct* values higher than 40 (as indicated in the negative controls) were considered negative for *Plasmodium* species. The amount of parasite DNA in a sample was quantified using the following equation: Parasite DNA (per/μL) = [2 ^E×(40−*Ct*sample)^/10]; where *Ct* is the threshold cycle of the sample and E is the amplification efficiency.

### 2.6. Duffy Blood Group Genotyping

A 1100-bp fragment of the human *DARC* (Duffy antigen receptor for chemokines; FY) gene that encompasses the GATA-1 transcription factor-binding site of the gene promoter was initially amplified using published primers [13]. In addition, another set of primers for the human *DARC* were designed using Primer 3plus software (Macrogen In., Seoul, Korea). The first PCR includes an upstream 5’ GTGGGGTAAGGCTTCCTGAT 3’ and downstream 5’CAGAGCTGCGAGTGCTACCT 3’ primers which amplify 998 bp of exon 2 of the *DARC* gene (125 G→A). The second PCR amplified a 638 bp fragment which includes the GATA-1 transcription factor binding site of the *FY* gene (−33T→C) using an upstream primer 5’ GGATGGAGGAGCAGTGAGAG 3’ and downstream primer 5’CAAAGGGAGGGACACAAGAG 3’. The PCR reaction contained 20 μL DreamTaq PCR Mastermix, 1 μL of DNA template, and 0.5 μL of each primer. PCR conditions were 94 °C for 2 minutes, followed by 35 cycles of 94 °C for 20 seconds, 58 °C for 30 seconds, and 68 °C for 60 seconds, followed by a 4 minute extension. PCR products were sequenced and the chromatograms were visually inspected to determine the *Fy* genotypes (see the Duffy blood group nomenclature in [14]. The PCR conditions were as follows: Initial denaturation at 95 °C for 3 minutes, followed by 35 cycles of 94 °C for 30 seconds, 55 °C for 1 min and 72 °C for 1 min, with a final extension at 72°C for 10 minutes. PCR products were purified and sequenced using both 5’ and 3’ primers on a 3730 DNA Analyzer (Macrogen Inc., Amsterdam, The Netherlands)

### 2.7. Statistical Analyses

Descriptive and inferential statistics were performed using IBM SPSS version 21. Fisher’s exact test was used to compare proportions of Duffy genotypes in relation to infection. Odds ratios were performed to measure the strength of association between *P. vivax* infection and Duffy phenotypes. Gene sequences were visualized using FinchTV software version 14.0 [20] nucleotide BLAST. Multiple sequence alignment was done using BioEdit Sequence Alignment Editor software version 7.2. All reference sequences were retrieved from the NCBI web site [21] using the Basic Local Alignment Search Tool (BLAST).

## 3. Results

### 3.1. Distribution of Duffy Phenotypes among P. vivax Infections

Out of the 992 microscopy positive samples, 186 were confirmed as *P. vivax* mono-infection and an additional four were identified as mixed infections of *P. vivax* and *P. falciparum* based on nested PCR. We further confirmed *P. vivax* infections by DNA sequencing the *P. vivax* 18S rRNA gene in six monoclonal infected samples (accession number MF540769-MF540774) when compared with the reference *P. vivax* Sal strain (accession number U07367.1). The mean parasite count in the positive samples was 5453.7 parasite/µL (95% confidence interval 2896 to 8010).

Duffy genotyping was performed in 190 *P. vivax* positive patients (as confirmed by both microscopy and PCR), and in 67 healthy non-malaria infected individuals with matched age and gender. Among the 190 PCR-confirmed *P. vivax* samples (186 mono-infections and 4 mixed infections), 129 samples (67.9%) were Fy(a-b+), whereas Fy(a+b-) and Fy(a-b-) only accounted for 14.2% and 17.9%, respectively, of the samples (Table 1). No Fy(a+b+) was detected among the samples. The proportion of the three Duffy phenotypes in healthy non-malaria individuals was significantly different from that in *P. vivax* infected patients. Among the 67 healthy non-malaria infected individuals, 45 (67.1%) were Fy(a-b-), followed by Fy(a-b+) (29.9%). Fy(a+b-) was the least prevalent with only 3% of the samples (Table 1). The Fy(a-b+) and Fy(a+b-) phenotypes were shown to be significantly higher in the *P. vivax*-infected patients than the healthy individuals (*p* = 0.01 and *p* < 0.01, respectively). By contrast, a significantly lower proportion of *P. vivax* infection was observed in Duffy-negative Fy(a-b-) individuals (*p* < 0.01; Table 1).

For all three sites (New Halfa, Khartoum, and River Nile), Fy(a-b+) was the highest prevalent phenotype. In New Halfa, 25 out of 40 (62.5%) *P. vivax* samples were Fy(a-b+) though other phenotypes, including (Fy(a+b-) and Fy(a-b-), were also detected with a smaller proportion (Figure 1). The prevalence of infection in Khartoum and River Nile showed a similar trend, Fy(a-b+) was 80.2% and 46.9%, respectively (Figure 1, Table 1, Appendix A).

### 3.2. Parasite Density and Demographic Features in Duffy-Negatives

Parasite density of a subset of 77 *P. vivax* samples representing the three Duffy phenotypes was estimated by qPCR assay. A significant difference was observed in the parasite density among the Duffy phenotypes (Figure 2). Individuals of Fy(a-b-) had the lowest amount of *P. vivax* parasites (mean log parasite density = 1.35 ± 0.63 µL^−1^) compared to individuals of Fy(a+b-) (mean log parasite density = 3.67 ± 1.02 µL^−1^) and Fy(a-b+) (mean log parasite density = 3.96 ± 0.83 µL^−1^) phenotypes (*p* < 0.001; Figure 2). No significant difference was detected between the Fy(a+b-) and Fy(a-b+) individuals (*p* = 0.48), despite a relatively small number of Fy(a+b-) individuals.

For a subset of 77 *P. vivax* samples, the average age of Fy(a-b-) individuals was similar to those of Fy(a-b+), but significantly younger than those of the Fy(a+b-) phenotype (Figure 3A, Appendix A). Despite a small sample size, the numbers of males and females were almost similar among the Fy(a-b-) and Fy(a+b-) individuals, whereas a larger number of males than females was observed among the Fy(a-b+) individuals (Figure 3B, Appendix A).

## 4. Discussion

This study was carried out in three *P. vivax* malaria endemic areas (Khartoum, New Halfa, and River Nile) in Sudan where Duffy positive and Duffy-negative individuals live side-by-side [22]. While a large number of Fy(a-b+) individuals were detected among the *P. vivax* infections, approximately 17% of these infections were Fy(a-b-) Duffy-negatives in our study sites. In the past, individuals with the Duffy-negative phenotype (Fy(a-b-)) were thought to confer resistance to *P. vivax* infection [9]. However, recently, in many parts of Africa, *P. vivax* infection was reported in individuals with the Duffy-negative phenotype [12,15,23,24,25]. In this study, we found a relatively high percentage (17.4%; 34/190) of *P. vivax*-infected patients who were Duffy-negative, much higher than that reported in Madagascar, where 8.8% of Duffy-negative hosts were *P. vivax* positive [13]. This finding confirms our previous study in central Sudan [4] and supports the hypothesis that Duffy-negativity is not fully protective against *P. vivax* infection [13].

Among the three study sites, River Nile had a higher proportion of Duffy-negative individuals (38.8%; 19/49) in comparison to Khartoum (7.9%; 8/101) and New Halfa (17.5%; 7/40). It is unclear if this contrasting pattern is related to the ethnicity difference among Sudanese in the three study sites. In this study, we did not detect any unique demographic features among these the Duffy-negatives who were infected with *P. vivax*. The epidemiological factors related to *P. vivax* infections in Duffy-negatives merit further investigation.

Compared to the *P. vivax* infected individuals, the majority of the healthy uninfected individuals were Duffy-negative Fy(a-b-) whereas a majority of the *P. vivax*-infected individuals were Duffy positive Fy(a-b+). This pattern agrees with the expectation that homozygous (Fy(a-b-)) individuals are less susceptible to *P. vivax* infection and corroborates with the previous reports from Ethiopia [26]. The observation of a small proportion of Fy(a+b-) individuals infected with *P. vivax* compared to Fy(a-b+) individuals suggest that these individuals could have a relatively low susceptibility to *P. vivax* infection, consistent with previous observations in Brazil and Papua New Guinea [27,28], or it may be due to the low frequency of this phenotype in our population. Future investigation is needed with a larger number of Fy(a+b-) individuals to test this hypothesis.

The observation of a high *P. vivax* infection among homozygote Duffy-negative (Fy(a-b-)) in this study suggests that the association between *P. vivax* and the Duffy antigens is more complex than habitually described. Further, the observation of lower parasitemia in Duffy-negatives corroborates previous findings in Ethiopia [24], supporting the hypothesis that the infectivity of the parasite to human red blood cells is reduced in the absence of or with reduced Duffy antigen expression. The parasite could have evolved an alternative red cell invasion pathway that is independent of Duffy antigen. This hypothesis merits further experimental studies to elucidate the molecular pathogenesis of *P. vivax* infection in the absence of the Duffy receptor and in-depth analyses of the level of Duffy antigen expression in different Duffy phenotypes.

In summary, this study indicates a relatively high rate of *P. vivax* malaria in Duffy-negative (Fy(a-b-)) individuals, particularly at the River Nile site in northeastern Sudan. While our data lends support to low parasitemia in the Duffy-negative Fy(a-b-) individuals in Sudan, the high prevalence may imply a Duffy-independent invasion pathway of the parasite or change in the host susceptibility, which can in turn drastically influence *P. vivax* distribution. *P. vivax* malaria could become a new epidemic or severe disease in East Africa given that the current antimalarial vaccine is developed based on the Duffy binding protein.

## Figures and Tables

**Figure 1 genes-10-00437-f001:**
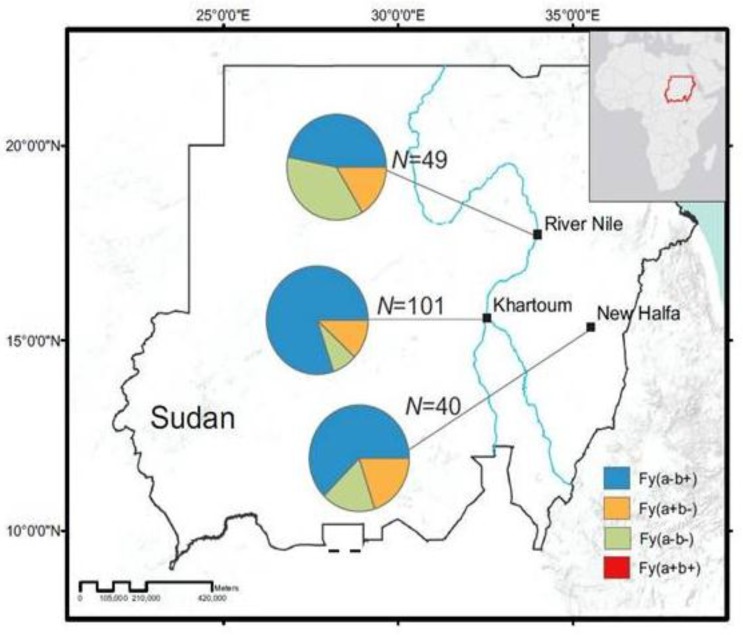
Proportion of Duffy phenotypes in *Plasmodium* positive patients among the three sites in River Nile, Khartoum, and New Halfa. The proportion of Fy(a-b-) was significantly higher in River Nile than the other two sites. In addition, Fy(a+b+) was not recorded in any of the healthy or *Plasmodium*-positive patients.

**Figure 2 genes-10-00437-f002:**
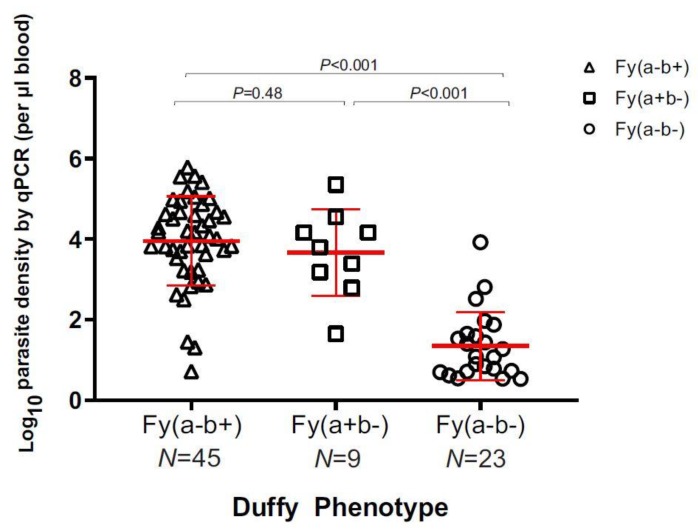
Estimated log parasite density per μL of blood based on Duffy phenotype. There was no significant difference between Fy(a-b+) and Fy(a+b-) phenotypes, but there was a significant difference between the Fy(a-b-) phenotype and the Fy(a-b+) and the Fy(a+b-) phenotypes (*p* < 0.001).

**Figure 3 genes-10-00437-f003:**
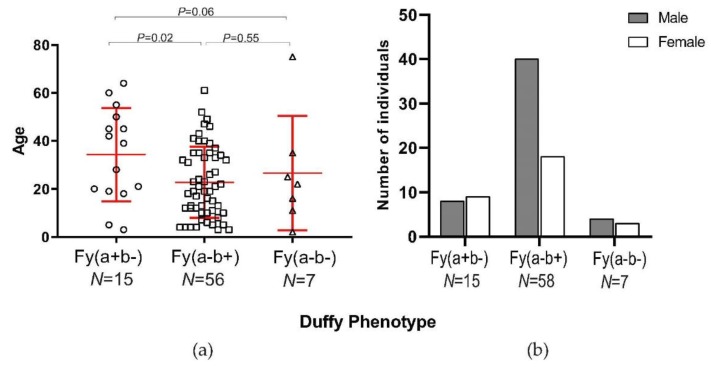
(**a**) Distribution of age of patient in years among the different phenotypes in *Plasmodium* positive patients; (**b**) proportion of male and female *Plasmodium*-positive patients between the different Duffy phenotypes. It is noted that there were two patient samples that we did not have their age data for. Thus, the sample size in Fy(a-b+) was different.

**Table 1 genes-10-00437-t001:** Proportion of the three Duffy phenotypes in *P. vivax* positive and negative healthy patients

Duffy Phenotype	*N*	Positive (*N* = 190)	Negative (*N* = 67)	*p* ^1^	OR (95%CI)^2^
Fy(a+b-)	29	27 (14.2%)	2 (3%)	0.0122	0.1858
Fy(a-b+)	149	129 (67.9%)	20 (29.9%)	0.0001	0.2012
Fy(a-b-)	77	34 (17.9%)	45 (67.1%)	0.0001	1.9567

^1^*p*-values were calculated by Fisher’s exact test at a significance level of 0.05.* Significant difference between the two groups after correction for multiple testing.^2^ Odd’s ratio with a 95% confidence interval.

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
