# Peer review of "Distribution of Duffy Phenotypes among Plasmodium vivax Infections in Sudan"

_genes, 2019, doi:10.3390/genes10060437_

Reviewer 1 Report

Summary:

This manuscript describe a study that assessed the presence of duffy negative expression in the blood of P. vivaxinfected individuals in Sudan. The authors state that several studies have reported that Negative Duffy expression on the surface of human red blood cells was a barrier for Plasmodium vivaxinfection in most Africans. However, due to recent data showing that P. vivaxcan infect Duffy negative individuals in several Central and East African countries, this study wanted to investigate the distribution of Duffy blood group phenotypes with regard to P. vivaxinfection and parasitemia in Sudan. They report that the Duffy Fy(a‐b+) phenotype was the most prevalent among the P. vivaxinfected patients, accounting for 67.9% of all infected phenotypes, while 17.9% of the homozygous Duffy negative individuals had P. vivax infection. The prevalence of infection in Fy(a‐b+) and Fy(a+b‐) phenotypes was shown to be significantly high than in duffy negative individuals. They observed P. vivaxinfection in Duffy negative individuals although the parasitemia  was low.

Comments:

This is very well written paper. The motivation behind the study was clearly stated and the methods to address the question were good.

There are a few typographical errors. Please review very carefully and correct them.

Author Response

Response to Reviewer 1 Comments

Point 1: “This is very well written paper. The motivation behind the study was clearly stated and the methods to address the question were good.”

Response 1: We thank you for your appreciation for our hard work of three years.

Point 2:There are a few typographical errors. Please review very carefully and correct them.”

Response 2: We have reviewed and corrected the typographical errors throughout the manuscript to our best.

Reviewer 2 Report

This article is relevant enough for the field of Plasmodium vivax research because it brings more light on the issue of P. vivax infecting duffy negative individuals and suggests the link between a Fy gene mutation (especially the b allele) and the increased possibility to infect individuals.

Nevertheless, there are minor corrections to do. In the abstract the number of P. vivax infected individuals change from 186 to 190 without any explanation.

Moreover, the Introduction is light and the references are old I can mention those of Cameroon, where there are fresh data available from 2017 published by Russo et al. which were not cited. Considering that P. vivax is a hot topic raising a lot of interest, I will suggest to the author to look around for more recent data in order to complete the one he has.

Lastly, it could be interesting to understand why only a subset of the P. vivax infected individuals underwent parasitemia quantification. this data which is of great importance should have been performed for all P. vivax  infected individuals.

Author Response

Response to Reviewer 2 Comments

Point 1: “Nevertheless, there are minor corrections to do. In the abstract the number of P. vivax infected individuals change from 186 to 190 without any explanation.”

Response 1: We apologize for this confusion. There were 186 patients which were tested as P. vivax positive mono-infection. An additional 4 patients were tested positive as mixed infections with P. vivax and P. falciparum. The combination of these two groups brings the total of patients with P. vivax to 190. We have corrected this in the abstract and Result.

Point 2:Moreover, the Introduction is light and the references are old I can mention those of Cameroon, where there are fresh data available from 2017 published by Russo et al. which were not cited. Considering that P. vivax is a hot topic raising a lot of interest, I will suggest to the author to look around for more recent data in order to complete the one he has.”

Response 2: We appreciate this comment and have made changes to the introduction with more recent references cited.

Point 3:Lastly, it could be interesting to understand why only a subset of the P. vivax infected individuals underwent parasitemia quantification. this data which is of great importance should have been performed for all P. vivax  infected individuals.”

Response 3: We do recognize the importance of including all samples for PCR quantification. Unfortunately, due to limitations in funding/resources and availability of samples, we did our best effort and completed qPCR assay on half of the positive samples.

Reviewer 3 Report

Title: Distribution of Duffy phenotypes among Plasmodium vivax infections in Sudan”

By Albsheer et al.

 In this study, the authors investigated the distribution of P. vivax infections in three different regions of Sudan. They observed 17.49% of P. vivax positive infections in Duffy negative individuals Fy(a-b‐), while 67.9% and 14.2% of P. vivax infections were observed in Fy(a-b+) and Fy(a+b‐) individuals respectively. Their results confirm existing reports of P. vivax infections in Duffy negative individuals in many countries of Central and East Africa.

Minor corrections:

1. Table 1. Proportion of P. vivax infections in Fy(a-b‐) individuals is 17.9% and not 17.49%.

2. Fig. 3. Cross check numbers of P. vivax positive Fy(a-b+) in Fig 3a and 3b.

3. Author should review article for Englsih

Author Response

Response to Reviewer 3 Comments

Point 1: “Table 1. Proportion of P. vivax infections in Fy(a-b) individuals is 17.9% and not 17.49%.”

Response 1: The correction has been made both in the table as well as within the text of the paper

Point 2:Fig. 3. Cross check numbers of P. vivax positive Fy(a-b+) in Fig 3a and 3b.”

Response 2: We have cross-checked the numbers and the data presented in Fig 3 is accurate. There were two patient samples that we did not have their age data. This explains the sample size difference in Fy(a-b+).

Point 3:Author should review article for English”

Response 3: We have done our best to review any spelling or grammar issues within the paper.